# Eco-Friendly Methods for Extraction and Modification of Cellulose: An Overview

**DOI:** 10.3390/polym15143138

**Published:** 2023-07-24

**Authors:** Solange Magalhães, Catarina Fernandes, Jorge F. S. Pedrosa, Luís Alves, Bruno Medronho, Paulo J. T. Ferreira, Maria da Graça Rasteiro

**Affiliations:** 1University of Coimbra, CIEPQPF, Department of Chemical Engineering, 3030-790 Coimbra, Portugal; solangemagalhaes@eq.uc.pt (S.M.); csfernandes@uc.pt (C.F.); jpedrosa@uc.pt (J.F.S.P.); paulo@eq.uc.pt (P.J.T.F.); mgr@eq.uc.pt (M.d.G.R.); 2MED—Mediterranean Institute for Agriculture, Environment and Development, CHANGE—Global Change and Sustainability Institute, Universidade do Algarve, Faculdade de Ciências e Tecnologia, Campus de Gambelas, Ed. 8, 8005-139 Faro, Portugal; bfmedronho@ualg.pt; 3FSCN, Surface and Colloid Engineering, Mid Sweden University, SE-851 70 Sundsvall, Sweden

**Keywords:** cellulose, functionalization, cationization, anionization, hydrophobicity, cellulose extraction

## Abstract

Cellulose is the most abundant renewable polymer on Earth and can be obtained from several different sources, such as trees, grass, or biomass residues. However, one of the issues is that not all the fractionation processes are eco-friendly and are essentially based on cooking the lignocellulose feedstock in a harsh chemical mixture, such as NaOH + Na_2_S, and water, to break loose fibers. In the last few years, new sustainable fractionation processes have been developed that enable the obtaining of cellulose fibers in a more eco-friendly way. As a raw material, cellulose’s use is widely known and established in many areas. Additionally, its products/derivatives are recognized to have a far better environmental impact than fossil-based materials. Examples are textiles and packaging, where forest-based fibers may contribute to renewable and biodegradable substitutes for common synthetic materials and plastics. In this review, some of the main structural characteristics and properties of cellulose, recent green extraction methods/strategies, chemical modification, and applications of cellulose derivatives are discussed.

## 1. Background

The rapid population growth and rise of globalization have been followed by the depletion of fossil fuel reserves, increasing health/environmental concerns. These have led researchers worldwide to look for new renewable resources for a more sustainable future. Cellulose, as the main component of plants (e.g., trees, grasses, agriculture residues, etc.), is the most abundant biopolymer on earth. Due to its high availability, low cost (especially from lignocellulosic residues), biodegradability, appealing physical properties, and chemical reactivity (potential for functionalization), cellulose has been receiving great attention from the research community over the last decades. Recent renewed interest has arisen due to its potential use as a renewable energy platform and for the development of cellulose-based materials.

A crucial contribution to the global climate challenge comes from forests and forest-based products, which store ca. 447 million tons of CO_2_ [1]. In fact, it is possible to prevent 410 million tons of carbon emissions per year by substituting fossil-based materials and fossil energy [2]. Thus, cellulose appears as a very appealing feedstock that can be used for the production of valuable chemicals through a variety of designed processing technologies without competing with the food industry or threatening the world’s food supply (contrary to other resources, such as starch) [3]. For example, glucose is a versatile precursor to obtain valuable chemicals such as biodegradable plastics and ethanol [3].

However, cellulose processing is challenging due to some important disadvantages, such as its insolubility in water and in most common solvents and its low resistance against microbial attacks [4]. Furthermore, cellulose can be chemically modified by substitution of its native hydroxyl groups with functional groups, such as specific acids, chlorides, and oxides, to address less favorable properties or to develop new desired characteristics. Over the last decades, many scientists have dedicated their research to the development of innovative ways to improve and tune cellulose properties and grant them new functionalities. These strategies typically involve cellulose derivatization by incorporation of cationic, anionic, or hydrophobic functional groups in its chain [5,6,7], thus broadening cellulose properties and applications.

In the current review, we explore the (1) fundamental structural characteristics and properties of cellulose from various sources; (2) sustainable extraction processes; (3) chemical modifications used in the preparation of cellulose derivatives; and (4) various applications for cellulose derivatives. Special attention is given to recent sustainable strategies engaging extraction, dissolution, and modification of cellulose, with a particular focus on the utilization of deep eutectic solvents. In the literature, it is possible to find different reviews dealing with cellulose derivatives and their applications. However, most of them are focused on specific applications, such as biomedical applications [4], food packaging [8], or wastewater treatment [9]. The present review also aims at bringing together the most recent developments regarding cellulose modification and the application of such cellulose derivatives, covering a wide range of modification techniques with a focus on sustainable chemistry.

## 2. Sources of Cellulose

Cellulose is generated at the plasma membrane in the form of paracrystalline microfibrils [10]. The hierarchical organization of cellulose in plants is illustrated in Figure 1. Individual microfibrils form cellulose fibrils that are located on the cell walls of plants. Cellulose is the main structural component of plants, and it is responsible for its structural support, providing strength and stability to the plant cell walls [3].

Cellulose can be obtained from various biomass sources, such as hardwoods (e.g., poplar wood, acacia, or eucalyptus wood), softwoods (e.g., pine wood or spruce wood), forestry residues, agricultural wastes, or grasses. Most properties of the obtained cellulose are strongly dependent on the source of biomass; one of these properties is the molecular weight, which has deep effects on cellulose application and processability. For example, hardwood raw materials typically present degrees of polymerization (DPs) ranging from ca. 1400–1790 [12], but it is also possible to find DPs of 2200–2300 [13,14]. On the other hand, softwood raw materials usually present DPs in the range of 2100 to 4750. Other lignocellulosic biomasses, such as grass, present DP values in the range of 1600–1900 [15].

The extraction process used to obtain cellulose also has an impact on the DP of the recovered cellulose. For instance, the sulfite process leads to less depolymerization than the kraft process [12,16]. Another common cellulose raw material, cotton, presents a DP of ca. 2150 [12]. Additionally, it is known that high-DP cellulose is more difficult to dissolve compared with cellulose of lower DP [17,18]. It is important to note that cellulose is also produced in nature by some bacteria and can be found in marine tunicates [19]. Some examples of cellulose sources and their relative content are listed in Table 1.

In addition to cellulose, lignocellulosic biomass is composed of two other structural polymers (hemicellulose and lignin), along with other minor compounds such as proteins and fatty acids [20,21]. Cellulose can be extracted and isolated from the other components in the raw material (typically in the form of cellulose fibers) by removing hemicellulose, lignin, and other impurities. The properties of the extracted cellulose fibers depend on their chemical composition, which, in turn, varies according to the source and even depending on the part of the plant where it comes from, as well as on the applied separation process.

**Table 1 polymers-15-03138-t001:** Various sources of cellulose and relative amounts (adapted from reference [3]).

Lignocellulose Biomass	Cellulose Source	Cellulose (%)	Ref.
Hardwood	Poplar	35–50.0	[21,22,23]
	Oak	40.4	[24]
	Eucalyptus	40–45.0	[25,26,27]
	Acacia	40–45.0	[28]
Softwood	Pine	42.0–50.0	[29,30]
	Douglas fir	40.0–50.0	[31,32]
	Spruce	45.5	[33]
Agriculture waste	Wheat straw	35.0–39.0	[34]
	Barley hull	34.0	[35]
	Barley straw	36.0–43.0	[36,37]
	Rice straw	29.2–34.7	[38,39,40]
	Rice husks	28.7–35.6	[41]
	Oat straw	31.0–35.0	[42]
	Corn cobs	33.7–41.2	[43]
	Corn stalks	35.0–39.6	[44]
	Sugarcane bagasse	25.0–45.0	[45]
	Sorghum straw	32.0–35.0	[46]
Grasses	Grasses	25.0–40.0	[47]
	Switchgrass	35.0–40.0	[48]

Considering what is referred to above, the choice of the cellulose source will depend on the desired properties and application, its availability, and economic purposes. Nowadays, most of the cellulose fibers used worldwide are extracted from wood. Nonetheless, wood is not widely available in some regions, and there is also a competing interest among several industries related to construction, furniture, pulp and paper, and the burning of wood for energy harvesting. Thus, it can be challenging to supply the required quantities of wood at reasonable prices to all sectors [49]. This encourages the use of other non-woody sources, such as herbaceous or aquatic plants, grasses, crops, and their by-products, for a variety of applications. These non-woody plants generally contain less lignin than wood, making the bleaching methods less demanding in terms of both chemicals and energy consumption.

## 3. Green Methods for Cellulose Extraction

There are various lignocellulosic biomass fractionation processes that allow the separation and isolation of the components; the choice of the most efficient method depends on the target polymer, source, and desired properties of the final product. Usually, cellulose is obtained by dissolving lignin and hemicellulose, along with low-molecular-weight compounds.

Conventional methods for biomass fractionation, such as those used in the pulp and paper industry (e.g., kraft cooking), are very efficient for the extraction of cellulose, but the nature of the solvents used and the harsh treatment conditions employed have led the scientific community to search for new environmentally friendly alternatives. The use of green solvent systems, such as ionic liquids (ILs) and deep eutectic solvents (DES), has been reported for biomass fractionation and demonstrated to be very promising systems (Figure 2), not only because of their high efficiency and selectivity but also due to their inherent advantageous properties, such as low environmental impact, low toxicity, biodegradability, good stability, and easy recycling routes [20,50,51,52].

ILs, first reported by Paul Walden in 1914, are known as “molten salts” because they present a low melting point, usually below 100 °C [53], and are very promising solvents for the dissolution and/or isolation of large biomolecules, such as cellulose and lignin [54,55]. The ILs contain organic cations, usually quaternary aromatic or aliphatic ammonium ions. Alkylated phosphonium and, occasionally, sulfonium cations can also be included in the IL chemistry [56]. The IL anion plays an important role in the IL’s ability to dissolve cellulose. Suitable ILs identified to date contain anions such as chloride, carboxylates, dialkyl phosphates, dialkyl and trialkylphosphonates, and amino acid anions [57]. The dissolving ability of these relevant ILs has been typically attributed to strong hydrogen-bonding interactions between the anions and equatorial hydroxyl groups present on the cellulose molecules [55,58]. Nevertheless, since the cations are typically bulky with delocalized charge, this has been argued to also favor the dissolution of amphiphilic-like molecules, such as cellulose [18]. Biomass deconstruction greatly depends on the ability of the IL to establish intermolecular interactions with lignocellulosic components, and several ILs have been reported to selectively dissolve cellulose [50,59]. The strength of the interactions between cellulose and the IL ions can be tuned by modifying the IL composition [60]. However, IL-based processes have been, so far, mainly applicable for lab-scale experiments. Apart from the questionable “green features” of ILs, their relatively high viscosity and high cost of production and purification still hinder pilot and industrial-scale trials. Therefore, the development of novel systems capable of efficiently and selectively extracting the main biopolymers present in biomass is highly desirable, particularly if greener and more environmentally friendly systems are prioritized.

In this context, DES have emerged as promising solvent systems due to their greener profile and high efficiency for biomass fractionation [50,61]. The first DES was synthesized by Abbott et al. in 2004 [62] and was formed by a mixture of Bronsted or Lewis acids (hydrogen bond donors (HBDs)) combined with quaternary ammonium salts (e.g., Choline chloride (ChCl)) (hydrogen bond acceptors (HBAs)) [63,64,65]. When mixed at a certain molar ratio, the melting point of the mixture becomes significantly lower than that of the original components [61]. The physical properties of DES, such as low melting point and volatility, high thermal stability, conductivity, and surface tension, are similar to those of room-temperature ILs [62]. These physicochemical properties can be further tuned by changing the HBD or HBA composition, which will consequently affect their performance as extraction media. DES systems are easy to prepare in a pure state, do not require the presence of any other solvent, and produce no waste. DES can be formed by natural bio-sourced cations and anions, such as those obtained from natural organic acids, amino acids, non-nutritive sweeteners, or natural compounds like choline or betaine, thus making these systems low-cost, non-toxic, and highly biodegradable [66]. It is clear that DESs offer several advantages over conventional solvents, and, recently, huge interest has been generated regarding their application in biorefineries [67]. DESs have been reported to selectively dissolve and extract high-quality lignin with ca. 90% purity and a yield of nearly 60% (*w*/*w*) of the total lignin present in different sources, such as corn straw [68]. Systems composed of choline chloride (ChCl) and lactic acid [69], ChCl and monoetanolamine [67], and ChCl and levulinic acid [70] are particularly efficient in biomass pretreatment. These DES are claimed to promote proton-catalyzed cleavage of various chemical linkages (e.g., ether and ester bonds) in the lignin-carbohydrate complex and in lignin molecules. Generally, carboxylic acid-based DESs exhibit stronger performance in lignin fractionation than those containing other functionalities, regardless of the HBAs used [71]. Despite their favorable physicochemical properties and performance, DESs are still not widely used. These are relatively new systems in biomass processing, and thus more research is needed to validate their full potential and support their application at a larger scale [72].

## 4. Cellulose Structure

Cellulose is a high-molecular-weight linear homopolymer composed of D-anhydroglucopyranose units (AGU) connected by β(1–4)-glycosidic bonds [62]. Each AGU monomer is rotated relative to its neighbor by 180° around the chain axis, forming a disaccharide unit known as cellobiose (Figure 3).

As shown in Figure 3, each AGU has six carbon atoms and three hydroxyl groups covalently linked to the carbon atoms at the C2, C3, and C6 positions. These hydroxyl groups can undergo typical reactions with primary and secondary alcohols. As presented in Figure 3, the terminal monomers of the cellulose polymeric chain can be divided into two types: (i) a reducing end, in the form of a hemiacetal, at the C1 position, and (ii) a non-reducing end, with a free hydroxyl group, at the C4 position [61,62].

The reactivity of the hydroxyl groups combined with their tendency to establish hydrogen bonds is responsible for some of the characteristics of cellulose, such as its highly cohesive nature and remarkable mechanical features [63]. Through van der Waals forces and intra- and intermolecular hydrogen bonds, the cellulose molecules organize into elementary fibrils that consist of tightly packed and ordered regions (known as crystallites) and less ordered and amorphous regions, as schematically represented in Figure 4. All the above-mentioned interactions contribute to the insolubility profile of cellulose in water and in most of the common solvents.

The crystalline fraction typically ranges between 40% and 70% (*w*/*w*) of the total cellulose fiber, and it is very dependent on the cellulose source and extraction conditions [66]. As the diffusion phenomena are facilitated in the disordered regions, a notable ease of solvent penetration into cellulose fibers can be observed. Solvent diffusion into the less-ordered (amorphous) regions can induce the swelling of the fiber structure, further increasing its accessibility. The swelling effect can be induced by bases, acids, salts, and some organic solvents [73,74]. These swelling agents can penetrate into the inner core of the fibers and interfere with the hydrogen bonds and van der Waals forces, disrupting the fibrillar aggregates, loosening the structure, and thus making available additional surface hydroxyl groups [64].

In general, cellulose derivatives result from the non-homogeneous substitution of the hydroxyl groups in each AGU unit by other functional groups. Due to easier chemical accessibility, cellulose modification is expected to occur preferentially in the amorphous regions and, if allowed, later in the crystalline regions [64].

## 5. Cellulose Reactivity

As mentioned, cellulose has the capacity to participate in different chemical reactions due to the three hydroxyl groups in each AGU. Under heterogeneous conditions, the reactivity of the hydroxyl groups can be affected by: (1) their inherent chemical reactivity; (2) steric effects that may arise from the reacting agent; and (3) steric effects that are driven by the supramolecular structure of cellulose [75]. In most cases, the hydroxyl groups at the C2 and C3 positions behave as secondary alcohols, while the hydroxyl group located at C6 acts as a primary alcohol. The average number of OH groups in each AGU that have been substituted is known as the degree of substitution (DS) [76]. For example, if all three hydroxyl groups are substituted, DS is 3.0.

One important class of reaction in cellulose is esterification. In this respect, it has been found that the OH at the C6 position is more prone to react than the OHs at the other positions. Moreover, the OH at the C2 position reacts twice as fast as the OH at the C3 position in esterification reactions [77]. In comparison with the other two secondary hydroxyl groups, the primary hydroxyl group at C6 has an axis of free rotation around the C5–C6 bond, which leads to a more reactive behavior. Nevertheless, the reactivity of this primary alcohol depends on the oxidation conditions, type of oxidant, and pH of the medium [78].

Despite the favorable presence of reactive OH groups, reactions involving cellulose are typically not easy, mainly because cellulose is highly heterogeneous in nature. As discussed above, different parts of its constituent fibrils display very different accessibilities to the same reagent (amorphous vs. crystalline domains) [79]. The accessibility of the cellulose fibers can be improved by treatments with distinct solvents that can disrupt the internal structure of the fibers and promote a swelling effect, or by the application of mechanical treatments such as grinding [80]. Among these pre-treatments, swelling is the most frequently used activation method for cellulose modification. Swelling agents generally penetrate the highly ordered regions and break/weaken interactions among the fibrils, loosening the internal structure. Depending on the cellulosic raw material and strength of the solvent, cellulose fibers can be completely solubilized, and, depending on the prevailing medium conditions, cellulose can behave as an acid, a base, or an amphoteric compound [81].

The efficiency of the activation process (swelling/dissolution) deeply influences the capacity to facilitate and control reactions with the three hydroxyl groups in each AGU [72]. It is important to note that cellulose swelling and dissolution are two distinct processes. However, in order for swelling or dissolution to occur, the chemical agents are required to penetrate and diffuse into the inner fibrillar core of the cellulose. In the swelling process, the overall structure of cellulose remains essentially intact, regardless of some significant physical changes and the increase in volume due to the uptake of the swelling agent. On the other hand, full dissolution of cellulose implies the transition from a heterogenous two-phase system to a homogenous one-phase system by disrupting the well-organized and complex supramolecular structure of cellulose. Depending on the cellulose properties and operation conditions, a given solvent can either act as a swelling agent or as a dissolving medium [82]. Regardless of the differences between the two processes, from a physicochemical point of view, they both occur by overcoming the intermolecular interactions responsible for the cohesion of the fibrillar structure [83]. It is also important to note that, due to the structural heterogeneity of most cellulose samples, both processes may occur simultaneously during the treatment, resulting in the partial dissolution of the material and a more or less swollen fraction [72]. This illustrates how complex these systems and processes are without being possible to adopt a simple and straightforward standard strategy.

## 6. Chemical Modification of Cellulose

Cellulose is a fascinating polymeric material and possesses several favorable features, but it also presents some drawbacks, such as its poor solubility in common solvents and its lack of thermoplasticity and antimicrobial properties. To overcome such limitations, controlled chemical modification of the cellulose structure is often a suitable strategy [84,85].

Dimensionally speaking, cellulose derivatives fall into two main categories: macromolecular cellulose derivatives and nanoscale particles [86]. Etherification reactions, in which organic species, such as ethyl and methyl units, react with accessible hydroxyl groups of cellulose, are frequently performed to produce water-soluble cellulose derivatives. Cellulose-based materials can be developed by chemical modification through surface functionalization or copolymerization of cellulose ethers, e.g., carboxymethyl cellulose (CMC), hydroxypropyl methylcellulose (HPMC), methylcellulose (MC), hydroxyethyl cellulose (HEC), ethyl cellulose (EC), and cellulose acetate (CA) (see Figure 5). Regarding the nanoscale cellulose particles, they can be nanofibrils (CNF), nanocrystals (CNC), and bacterial cellulose (BC), being possible to prepare bacterial cellulose nanocrystals by acid hydrolysis of BC (BCNC) [86].

To improve cellulose reactivity and allow further modifications, pretreatments are often performed to introduce more reactive groups into the cellulose structure. One example is the oxidation of cellulose to convert the hydroxyl groups into more reactive aldehyde groups that can then undergo other derivatizations. Regarding the oxidizing reagents used in cellulose chemistry, many can be enumerated, such as nitrogen oxides, alkali metal nitrites and nitrates, ozone, permanganates, and peroxides. These agents usually lead to reactions with low selectivity. However, cellulose oxidation with periodates presents a very high selectivity [87], while minimizing the degradation of the cellulose chains and keeping acceptable mechanical and morphological properties [88]. Periodic acid and its salts, periodates, are known as regioselective oxidation agents capable of converting vicinal diols, such as those of carbohydrates, to dialdehyde structures [89]. In this case, the periodate will induce the cleavage of the C2–C3 bond of the AGU, with the consequent formation of carbonyl groups at those positions, resulting in dialdehyde cellulose (DAC), as described in Figure 6 [90].

Several reaction parameters may influence the properties of the obtained DAC, such as the concentration of periodate (i.e., higher concentrations of periodate improve the formation of aldehyde groups and allow for the cellulose to have higher aldehyde contents), temperature, and reaction time. The pH effect is also an important parameter to control during the oxidation of cellulose. It was reported that, in acidic conditions (pH < 3), the hydrolysis of cellulose is enhanced, resulting in superior degradation of the fibers [91]. Usually, selective oxidation with periodate is applied as the first step, in which the cellulose’s crystalline structure is partially dismantled. Often, during such processes, the degree of polymerization is also observed to decrease [5].

Although periodate oxidants are toxic, environmentally harmful, and relatively expensive, their recycling and reuse may make the process more sustainable and feasible, both from environmental and economic perspectives [89]. Moreover, the highly reactive aldehyde groups allow several different further derivatizations, such as sulphonates by bisulfate addition, carboxylic acid derivatives through further oxidation, and imines from a reaction with an amine [92]. The DAC derivative has been shown to be biodegradable and biocompatible, which can be beneficial in many potential applications [93]. These DAC characteristics are suitable to produce environmentally friendly, green-high-end materials with potential applications as bio-flocculants, complexing agents, and super-adsorbents [5].

### 6.1. Cationization

Cellulose, usually obtained as cellulose fibers from wood sources, is typically negatively charged due to the ionization of the hydroxyl groups. The functionalization of cellulosic materials with cationic moieties has been a chosen strategy to confer affinity toward other negatively charged molecules/particles and expand the applicability of cellulose derivatives. Cationic celluloses have been applied as bio-based flocculant and/or adsorbent alternatives for water treatments [94,95]. Additionally, the cationic groups can potentially disrupt the negatively charged bacterial cell walls [96], thus broadening the application of cationic cellulose towards the biocide area [97,98].

Two main strategies are described in the literature for the cationization of cellulose. The first one involves the physical adsorption of cationic polymers into the cellulose surface [99], and the second approach relies on the chemical modification and grafting of cationic groups into the reactive sites of cellulose. The covalent functionalization can be further subdivided into direct cationization, when the cationic groups directly attach to the hydroxyl groups of cellulose [100], or indirect cationization, in which cellulose is first derivatized to enhance its reactivity (via the introduction of, for instance, carbonyl groups) and later the intermediate derivative further reacts with the cationizing agent [101].

Although cations from various atomic elements can be used for cationization (onium salts from elements of the 15th to 17th group of the periodic table, such as quaternary ammonium or phosphonium and tertiary sulphonium cations), most of the literature focuses on the use of nitrogen-derived compounds. Depending on the derivative, the charge can be pH-dependent, with the cationic group being formed due to the protonation of amines (primary, secondary, or tertiary) or heterocyclic compounds (pyridine and imidazole) under acidic conditions. On the other hand, quaternary ammonium derivates present a permanent pH-independent positive charge [102,103].

One common method for cellulose cationization is based on direct modification by dissolution of short-chain cellulose molecules in aqueous solutions (e.g., NaOH/urea, NaOH/thiourea, or LiOH/urea), which are pre-cooled to sub-zero temperatures, followed by cationization in a homogeneous medium with N-(3-chloro-2-hydroxypropyl)trimethylammonium chloride (CHPTAC). In this system, the reactive epoxy reagent is prepared in situ by reacting CHPTAC with alkali (Figure 7A). The epoxy reacts with the available hydroxyl groups of cellulose to form an ether linkage, resulting in cationized cellulose (Figure 7B). At the same time, an unavoidable competing hydrolysis reaction occurs, where 2,3-epoxy-trimethylammonium chloride (EPTAC) is converted to the nonreactive form 2,3-dihydroxypropyltrimethylammonium chloride (Figure 7C). This undesirable parallel hydrolysis reaction represents a major drawback to the economic feasibility of this process since it compromises the reaction efficiency [100].

An alternative approach considers the cationization of pre-modified cellulose by using, for example, DAC [104], CA, or HEC [5] as raw materials. Regarding the DAC-based approach, after cellulose conversion to DAC by oxidation with sodium metaperiodate (described in Section 6 and Figure 6), DAC reacts with the Girard’s reagent T (GT), forming a stable imine structure with cationic quaternary ammoniums (Figure 8) [105,106].

The direct cationization of wood cellulose fibers with CHPTAC and the indirect method with GT were both tested by Pedrosa et al. [107] as a pretreatment to produce micro/nanofibrillated cellulose by high pressure homogenization. The morphological analysis demonstrates that the samples subjected to sodium metaperiodate oxidation (opening of the anhydroglucose ring) suffered significant degradation of the cellulosic structure, leading to the formation of short fibrils and enhanced solubilization of the material. A DS of 0.36 resulted in the complete solubilization of the cellulose fibers. The cationization with CHPTAC allowed for longer fibrils that conferred a more cohesive 3D structure (forming a gel-like material at ca. 1% (*w*/*w*)). The solubilization of the fibrils was not detected.

Sirviö et al. [108] reported the cationization of cellulose using betaine hydrochloride as a cationic reagent, tosyl chloride as a coupling agent, and a DES based on triethylmethylammonium chloride (TEMA) and imidazole (molar ratio 1:2). The reaction conditions, such as temperature, amount of cellulose, and reagents, were evaluated, and the DS was observed to vary from 0.07 to 0.44. From a mechanistic point of view, the imidazole acts first as a catalyst by deprotonating the betaine carboxylic acid group (Figure 9). Furthermore, the deprotonated betaine reacts with tosyl chloride to form a mixed anhydride. At this stage, the imidazole works as an acid scavenger that neutralizes the hydrogen chloride formed as a by-product. Moreover, the oxygen atoms of the hydroxyl groups of lignocellulose react with the anhydride to form an intermediate species. The intermediate species is then deprotonated by imidazole, resulting in the formation of cellulose betaine ester as the main product and p-toluenesulfonic (tosylic) acid as a by-product. Tosylic acid is then neutralized by imidazole. During the cationization process, the tosylation of cellulose occurs as a side reaction, possibly due to the presence of basic imidazole that may catalyze the formation of tosyl cellulose. By decreasing the tosyl chloride content, the authors were able to limit the occurrence of undesired side reactions and decrease the chemical consumption without significantly compromising the cationic group content, thus improving the environmental impact and economic feasibility of the procedure [108].

Emam et al. [109] reported the cationization of viscose fibers via a two-step reaction with a quaternary ammonium salt. Initially, viscose fibers are activated by an alkalization step using sodium hydroxide. The authors suggested that the hydroxyl groups of cellulose fibers are deprotonated in the presence of NaOH, and cellulose fibrils become more accessible through swelling. In the second step, cationic cellulose is obtained by modification of the activated fibers with N-2-chloroethyl-N,N-diethylammonium chloride (CEDAC). Via solvolysis, CEDAC forms an aziridinium ion, which is prone to react with the deprotonated hydroxyl groups of cellulose and form the cationized cellulose derivative (Figure 10). As suggested by the nitrogen content analysis, the DS is observed to be dependent on the concentration of quaternary amine.

### 6.2. Anionization

Although cellulose anionization is not as well explored as cationization, there are some procedures described in the literature. For example, Rajalaxmi et al. [110] and Grenda et al. [5] studied the synthesis of water-soluble anionic lignocellulose by sulfonation of DAC. Anionic DAC (ADAC) was obtained by dispersing DAC in deionized water and reacting it with sodium metabisulfite [5] or sodium bisulfite [110] (Figure 11).

In Grenda et al. [5], after the anionization reaction with sodium metabisulfite, the resultant transparent solution was mixed with isopropanol to precipitate the soluble product. The authors observed that as the reaction time increased, greater homogeneity in the product was achieved. It was reasoned that the presence of lignin restricts the penetration of sodium metabisulfite into the lignocellulose dialdehyde, requiring longer reaction times to provide sufficient sulfonation in the final product. However, the sulfur content in ADACs revealed that too long reaction times did not necessarily translate into higher degrees of sulfonation because the product undergoes chemical degradation for reaction times longer than 72 h. The authors suggest that, for anionization of DAC with sodium metabisulfite at room temperature, the optimal reaction time ranges between ca. 34 h and 72 h.

Cao et al. [111] have synthesized carboxylated cellulose by reacting cellulose with different acidic DES (Figure 12). Cellulose and the DES of interest were initially ball milled at 500 rpm for 30 min and then mechanically stirred for 2 h at 800 rpm and 100 °C. After the reaction, the product was thoroughly washed with ethanol and freeze-dried. Several DES composed of choline chloride as HBA and five different carboxylic acids (i.e., citric acid, malic acid, oxalic acid, malonic acid, and succinic acid) as HBDs were evaluated. It was observed that the acid’s chain length, molecular size, and number of hydroxyl and carboxylic groups affect the carboxylation efficiency of the DES. The decrease in the chain length of the carboxylic acid and, consequently, the increase in acidity lead to higher carboxylation efficiencies. However, carboxylation with DES containing citric acid—which has stronger acidity and more carboxylic groups—is somehow sterically hindered due to its molecular size. The most favorable ratio between acidity and molecular size of oxalic acid resulted in the highest carboxylation efficiency. The addition of a small amount of water improves the fluidity and mass transfer rate of the DES, thereby increasing the carboxylation efficiency of cellulose. Moreover, it also allows an easier penetration of chloride ions from HBA into the cellulose, which eventually contributes to dismantling the crystalline structure while promoting efficient cellulose carboxylation. Overall, the DES composed of choline chloride and oxalic acid at a molar ratio of 1:5 and containing 10% (*w*/*w*) water was found to be the most promising mixture. The use of ball milling in the process reduces the cellulose particle size, increases the surface area of cellulose, promotes the interaction between DES and the cellulose molecules, and disrupts the crystalline structure, thus increasing the carboxylic content in the modified cellulose.

### 6.3. Hydrophobic Modification

Cellulose molecules possess a great number of hydroxyl groups, leading to fibers with a strong polarity and high water sorption capacity. However, most of the synthetic polymeric matrices are nonpolar, such as plastics (polyethylene or polypropylene are among the most common) [113]. Therefore, the interfacial compatibility between cellulose fibers and polymeric systems is rather poor. For example, the addition of unmodified cellulose fibers to polymer composites often leads to a significant reduction in impact strength due to the poor compatibility between hydrophilic fibers and the hydrophobic polymer matrix. Thus, it is important to ensure that the bonding or adhesion between fibers and the polymeric matrix is sufficiently high to enhance the interfacial compatibility, which governs the mechanical properties of the composite materials. This can be achieved by chemically modifying cellulose and introducing hydrophobic groups in the cellulose chain. For example, Vehvilainen et al. modified enzyme-treated cellulose in alkaline aqueous tert-butanol, using allyl glycidyl ether as the modifying reagent, to obtain 3-allyloxy-2-hydroxypropyl cellulose (Figure 13) [114]. The modification was performed in a homogeneous medium, and cellulose with a high degree of substitution could be attained. However, its application is still restricted due to the harsh alkaline conditions required and the demanding operational details.

Homogenous acetylation and carbanilation reactions of wood-based lignocellulosic materials in ILs have also been investigated, resulting in highly substituted lignocellulosic esters (Figure 14). A high DS of 92.6% and 89.7% (based on the amount of OH groups substituted in lignocellulosic material, determined by ^31^P NMR) can be achieved under mild conditions for acetylated and carbanilated wood, respectively [68]. The optimal conditions were found to be 2 h for both reactions and a temperature of 70 °C and 80 °C for acetylation and carbanilation reactions, respectively. It was also reported that the IL can be recycled and reused without negatively affecting the reaction efficiency. This is considered a promising approach for surface modification of cellulose if ILs become routinely applied beyond the laboratory scale.

The preparation of hydrophobically modified cellulose from renewable feedstocks (based on green chemistry principles) can be met using plant oils. Plant oils are triglycerides with hydrophobic long hydrocarbon chains, which have been exploited as sustainable alternatives to materials derived from non-renewable resources [115]. Yoo et al. have reported the surface hydrophobization of CNC with bio-derived fatty acids using aqueous lactic acid as a reactive solvent without affecting the structural morphology and crystallinity of the grafted CNCs [116]. Similarly, Wei et al. have investigated the chemical modification of CNCs using canola oil fatty acid methyl ester via a transesterification reaction, as schematically illustrated in Figure 15 [117]. This transesterification strategy can be employed to modify other cellulose nanomaterials with a high availability of OH groups on the surface. Shang et al. have grafted diisocyanate-functionalized castor oil onto the CNC surface to enhance its hydrophobicity [118].

Another widely used way to modify the surface of polysaccharides and make them more hydrophobic relies on the introduction of acrylates and methacrylates into the chain. Littunen et al. studied the filling of various acrylates and methacrylates as monomers through a free radical copolymerization initiated by ammonium cerium (IV) nitrate ((NH_4_)_2_Ce(NO_3_)_6_) [119]. Initiation occurs via a redox reaction as the cerium ion reacts with two adjacent hydroxyl groups on a cellulose chain, resulting in the formation of a radical on an open glucose ring (Figure 16).

An important advantage of this method is that the entire synthesis can be performed in an aqueous medium. The macrostructures formed by the grafted polymers ranged from a thin coating to a continuous matrix completely enveloping the fibrils. This type of modification can offer a simple way of improving the compatibility between lignocellulosic materials and synthetic polymers.

The introduction of siloxane groups into the cellulose structure is another suitable approach to enhancing the hydrophobicity of the molecule. Schuyten et al. introduced, almost 70 years ago, the first cellulose derivative, trimethylsilylcellulose (TMSC) [121,122,123]. The TMSC was synthesized through the reaction of cellulose with different organochlorosilanes in the presence of pyridine. The ^13^C NMR spectrum in Figure 17 reveals the typical fingerprint of the synthesized TMSC. Only the signals for the substituent (0.0–2.0 ppm) and the AGU (103.0–60.8 ppm) are found. The authors have obtained TMSC with different degrees of substitution in a controlled manner, despite the observed low solubility in some relevant organic solvents, such as in a toluene/ethanol (80/20) mixture [124].

Later, some improvements in the TMSC synthesis led to products soluble in organic solvents, such as chloroform, 1,1,1-trichloroethane, and o-xylene [125]. The disadvantage of the process is the use of chemicals that pose risks for the user and the environment, such as pyridine and chloroform. Generally, the synthesis of TMSC is composed of several steps, involving cellulose dissolution in a non-volatile solvent, such as N,N-dimethylacetamide with LiCl, derivatization in the homogenous phase, and phase separation of the obtained TMSC. The final product can be dissolved in a suitable organic solvent (e.g., tetrahydrofuran or toluene) [123]. The obtained silylated cellulose is highly hydrolyzable in the presence of water or other hydroxylated compounds.

From the examples discussed above, it is clear that lignocellulosic materials can be modified in many different ways. The choice of the most suitable method will depend on the desired properties of the cellulose derivatives and the application foreseen.

## 7. Applications of Cellulose Derivatives

Chemical modification of low-cost, naturally occurring raw materials, such as cellulose, is an important and promising route for the development of green value-added products. Cellulose derivatives are currently used in different areas, such as food formulations, coatings, or films with barrier properties [4,11,126,127] (Table 2). Studies for the development of adhesives and composites have also been conducted, such as the production of cement-based composites from cellulose-modified fibers. Tonoli et al. [128] evaluated the effect of cellulose modification on the microstructure and mechanical properties of fiber-cement composites. Surface modification of cellulose fibers was conducted with methacryloxypropyltri-methoxysilane (MPTS) and aminopropyltri-ethoxysilane (APTS). The composites reinforced with APTS-modified fibers presented a higher modulus of rupture than those reinforced with unmodified or MPTS-modified fibers. The elasticity was observed to increase for both modified pulps due to the increased fiber-to-matrix adhesion [128].

An edible cellulose-based film for probiotic entrapment was prepared by Singh et al. [127] using sodium CMC and HEC. The use of non-toxic citric acid as a natural crosslinker makes these materials acceptable in the food and medical fields due to their excellent biocompatibility and hydrophilicity [129,130]. The mechanical, swelling, and release properties can be tuned by controlling the HEC/CMC ratio and amount of crosslinker. For example, HEC-based films show a higher swelling capacity than those containing CMC. On the other hand, the CMC films presented the highest tensile strength. In another study, Alves et al. developed composite films using TEMPO-oxidized cellulose nanofibers (CNF) and minerals. It was proven that those films have potential applications in food packaging and printed electronics [131]. The authors concluded that the presence of negatively charged groups resulted in higher fibrillation and, consequently, films with improved transparency and good mechanical and barrier properties. It was also observed that the TEMPO CNF films did not face a depletion in properties with the introduction of minerals, contrary to films prepared with non-modified CNF.

In the biomedical area, the use of cellulose derivatives as controlled drug release systems is very appealing [4,132]. Nanocrystalline cellulose is a promising material for biomedical applications because of its excellent mechanical properties and biocompatible nature. In addition, its high aspect ratio building blocks may construct natural crystals or nanorod networks that are held together by hydrogen bonding and physical entanglements. Such a network could be even further mechanically reinforced by cross-linking the individual nanofibers [4].

There are numerous cellular species that can be cultured on nanocellulose biomaterials, such as hydrogels, electrospun nanofibers, sponges, composites, and membranes [133]. Among the sources of nanocellulose, bacterial nanocellulose is believed to be the most suitable choice for cell culture due to its high purity, porosity, biodegradability, and low toxicity [134]. Other cellulose derivatives were employed in biomedical areas. For example, Bianchi et al. [135] prepared hydrogels with wound dressing capability in association with β-cyclodextrin; Suliwarno et al. [136] developed hydrogel-based materials, formed by electron beam irradiation crosslinking of methyl cellulose, also with wound dressing potential; Niemczyk-Soczynska et al. [137] used methyl cellulose to produce hydrogels, by thermally induced crosslinking, to create scaffolds for tissue engineering; Pasqui et al. [138] prepared carboxymethyl cellulose—hydroxyapatite hybrid hydrogels for composite materials for bone tissue engineering applications; and Dai et al. [139] developed PEG–carboxy-methylcellulose nanoparticles hydrogels for injectable and thermosensitive drug delivery.

Fuller et al. [140] studied the removal of munition constituents from stormwater runoff with native and cationic cellulose. The cationization of cellulose with CHPTAC was revealed to greatly improve the removal of 3-nitro-1,2,4-triazol-5-one (NTO) from both artificial and real stormwater runoff. An increase in the concentration of CHPTAC from 38 to 225 g/L was also revealed to positively enhance the removal. It was also demonstrated that the cationic materials tend to buffer the pH of the solution towards circumneutral values. This advantageous effect enables the NTO removal to not be affected by the initial pH of the medium [140].

Cellulose derivatives are also reported as efficient flocculants, namely for the flocculation of pigments [5] and calcium carbonate in papermaking [141]. Grenda et al. prepared cationic cellulose-based polyelectrolytes (PELs) to act as flocculants for dye removal in colored effluents [104]. The bleached Eucalyptus kraft pulp fibers were modified in a two-step reaction: DAC was initially prepared by the oxidation of cellulose with sodium periodate, and then the cationic groups were introduced by reacting the aldehyde groups with Girard’s reagent T. Cationic cellulose-based PELs with different properties (e.g., lignin content and charge) were tested at different concentrations and pH for the removal of dyes, and the addition or absence of an inorganic agent (bentonite) was also evaluated. The authors concluded that the dual system, with the addition of the inorganic agent followed by the flocculant, results in a generally higher color removal. Natural-based flocculants provided similar or even better performance in comparison to synthetic polymers. This ability to efficiently flocculate dyes with different structures and charge densities is of great importance due to the increasing number of industrial sectors that use dyes in their daily processes, such as the textile, pharmaceutical, paper, and cosmetic industries.

In the textile industry, cellulose modification can also be applied to enhance dye uptake into textile fibers. A potentially environmentally friendly dyeing method using a cationization method in combination with mercerization was proposed by Fu et al. [142]. The cationization of cotton was performed with CHPTAC, and it was observed that the dyeing performance of the cationized cotton fabrics is enhanced with the increase in cationizing agent. The dye uptake was higher than 95%, and an improvement in both color depth and colorfastness properties was observed. This method also allows the use of lower concentrations of dyes and avoids the extensive use of salt required by conventional methods, thus showing that this procedure is more environmentally benign than the conventional ones using fiber-reactive dyes. Pereira et al. have also recently reported the cationization of regenerated wood pulp fibers with glycidyltrimethylammonium chloride (GTAC) to improve dye uptake. The results show that GTAC-modified cellulose exhibits higher dye exhaustion (89.3%) and dye fixation (80.6%) values than non-modified cellulose [143].

**Table 2 polymers-15-03138-t002:** Summary of some selected applications of cellulose derivatives.

Reference	Molecular Weight/Degree of Polymerization	Type of Derivatization	Degree of Substitution	Application of Cellulose Derivative
Tonoli et al. [128]	Not provided	Silane grafting—methacryloxypropyltri-methoxysilane (MPTS) and aminopropyltri-ethoxysilane (APTS)	Not provided	Fiber-cement composite reinforcement
Singh et al. [127]	Carboxymethyl cellulose (CMC)—250 kDa; Hydroxyethyl cellulose (HEC)—720 kDa	Cellulose etherification	CMC—0.80–0.85; HEC 2.5 mol/mol cellulose	Edible cellulose-based films for probiotic entrapment
Alves et al. [13]	Degree of polymerization of 381	Oxidation with NaOCl in the presence of catalytic amounts of TEMPO and NaBr	Carboxylic group content: 0.74 mmol/g	Cellulose-based composite films for food packaging or printed electronics
Fuller et al. [140]	Not provided	Cellulose cationization with CHPTAC	ca. 700 μmol/g	Cationized cellulosic sorbents for the removal of insensitive munition constituents
Chambin et al. [132]	Hydroxypropylmethyl cellulose (HPMC)—low MW; Ethyl cellulose (EC)—230 kDa	Cellulose etherification	HPMC—1.9 methoxy groups per anhydroglucose unit; EC—Ethoxyl content% (*w*/*w*)—48–49.5	Matrices (granules and tablets) for drug release
Bianchi et al. [135]	Hydroxypropylmethyl cellulose (HPMC)—medium MW—500 kDa	Cellulose etherification	Methoxy content: 19–24% (*w*/*w*); Hydroxypropyl content: 1–7% (*w*/*w*)	Hydrogels with wound dressing capability, in association with β-cyclodextrin
Suliwarno et al. [136]	Methyl cellulose (MC)—18–27 kDa	Cellulose etherification	DS: 1.4–2.0	Hydrogel-based material formed by electron beam irradiation crosslinking, with wound dressing capability
Niemczyk-Soczynska et al. [137]	Methyl cellulose (MC)—13–16 kDa	Cellulose etherification	DS: 1.8	Hydrogels based on thermally induced crosslinking to produce scaffolds for tissue engineering
Pasqui et al. [138]	Carboxymethyl cellulose (CMC)—700 kDa	Cellulose etherification	Degree of carboxymethylation of 95%	Carboxymethyl cellulose-hydroxyapatite hybrid hydrogels for composite materials for bone tissue engineering applications
Dai et al. [139]	Sodium carboxymethyl cellulose (NaCMC)—275 kDa	Cellulose etherification	DS: 0.82	PEG-carboxy-methylcellulose nanoparticle hydrogels for injectable and thermosensitive drug delivery
Grenda et al. [5]	Not provided	Periodate oxidation to form cellulose dialdehyde is followed by anionization with sodium metabisulfite (ADAC) or cationization with Girard’s T reagent (CDAC)	ADAC—Anionicity index (mmol/g): 4.17–4.90; CDAC—cationicity index (mmol/g): 2.84–3.56	Anionic and cationic pulp-based flocculants for application in effluent treatment from the textile industry
Pedrosa et al. [141]	Not provided	Cellulose dialdehyde prepared through periodate oxidation and cationized with Girard’s T reagent (CDAC) or direct cationization of cellulose with CHPTAC	Charge density (mmol/g) of CDAC: 0.23–3.44; Charge density of CHPTAC modified cellulose (mmol/g): 0.46–0.92	Flocculants for calcium carbonate in papermaking
Fu et al. [142]	Not provided	Direct cationization with CHPTAC	Not provided	Environmentally benign method for dyeing textiles as a substitute for reactive dyes
Pereira et al. [143]	Not provided	Direct cationization of cellulose with glycidyltrimethylammonium chloride (GTAC) in a water/THF mixture	DS: 0.13–0.33	Dye fixation and dye exhaustion lead to textiles with enhanced dye uptake

## 8. Concluding Remarks

Cellulose is the most abundant natural polymer on Earth. Due to its wide availability, it is a very promising raw material for the replacement of non-renewable feedstocks. To overcome some limitations of its application and expand its valorization and utilization, cellulose can be chemically modified to improve its chemical and/or physical properties. Numerous studies reporting the modification and application of cellulosic materials are available in the literature. In the present review, some of the most common chemical modifications of cellulose (i.e., cationic, anionic, and hydrophobic modifications) are briefly described, and selected applications of these derivatives are presented. Cellulose and its derivatives can be obtained from different sources, including biomass such as agroforestry residues, and are suitable for a wide range of applications. A brief review of possible fractionation procedures is also presented, including greener alternatives. By understanding cellulose structure and reactivity, it is possible to tune the properties of the resultant material, such as by modulating the hydrophilic/lipophilic balance, charge, or degree of polymerization, to obtain materials with improved performance for the intended application. The overall range of material applications for cellulose derivatives is virtually limitless.

In summary, cellulose appears to be a sustainable, environmentally friendly feedstock with valuable properties such as its biocompatibility, non-toxicity, and wide availability, and the study of new routes to improve its properties and applications is of great interest. The cellulose derivatives are suitable for the replacement of fossil-based products and are an important alternative to reduce the environmental problems derived from the use of petroleum-based materials and fuels. Indeed, cellulose and its derivatives can provide the biological, chemical, physical, and engineering communities with new opportunities for exciting advancements and discoveries.

## Figures and Tables

**Figure 1 polymers-15-03138-f001:**
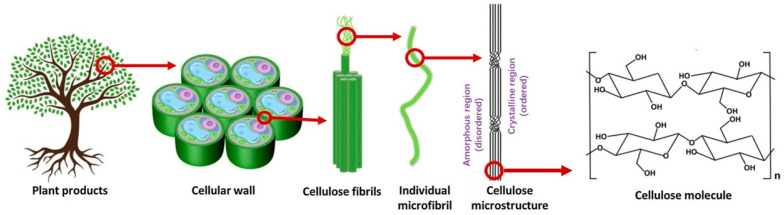
Illustration of the hierarchical organization of the cellulose chain leading to the formation of elementary fibrils, microfibrils, and the cellulose fibers from plant wood (adapted from [11] with permission from Elsevier).

**Figure 2 polymers-15-03138-f002:**
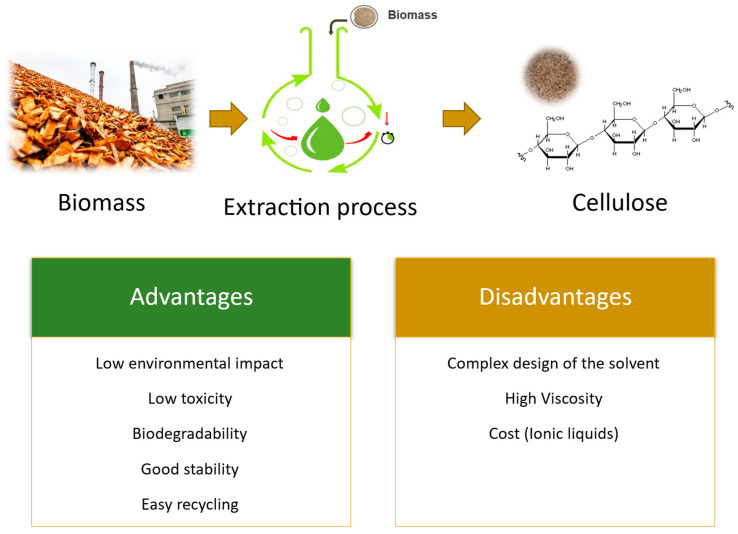
Advantages and disadvantages of green methods for cellulose extraction.

**Figure 3 polymers-15-03138-f003:**
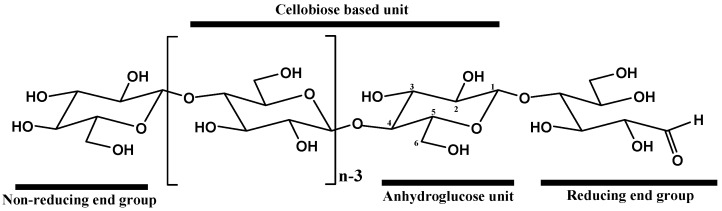
Molecular structure of cellulose showing the typical numbering of carbon atoms, the reducing end containing a hemiacetal group, and the non-reducing end with a free hydroxyl at the C4 position [66] (adapted from [66]).

**Figure 4 polymers-15-03138-f004:**
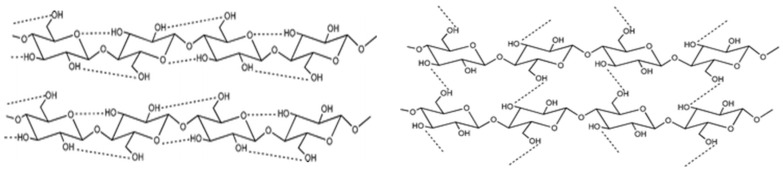
Intramolecular (**left**) and intermolecular (**right**) hydrogen bonding networks in cellulose molecules (from reference [64] with permission of the Royal Society of Chemistry).

**Figure 5 polymers-15-03138-f005:**
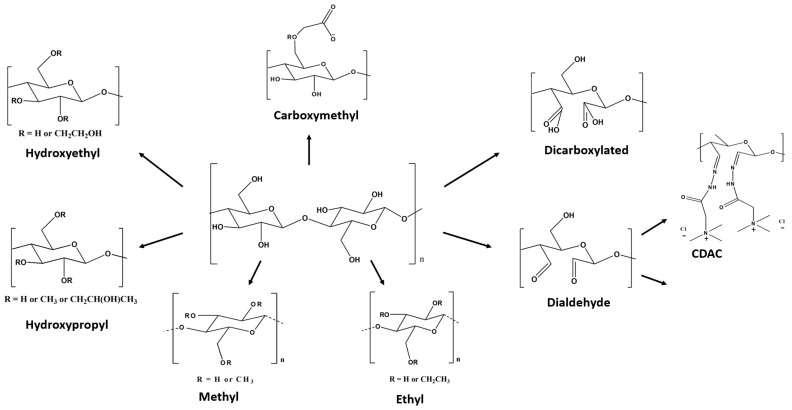
Schematic representation of the chemical structure of typical cellulose derivatives [86] (adapted from reference [86] with permission of the Royal Society of Chemistry).

**Figure 6 polymers-15-03138-f006:**
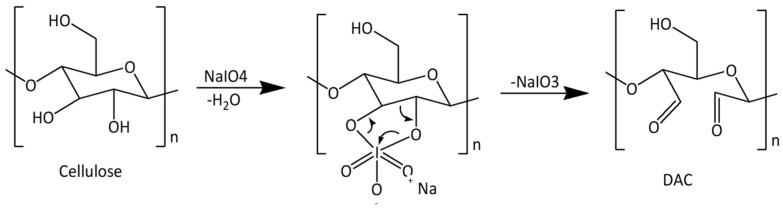
Reaction scheme of dialdehyde cellulose (DAC) synthesis [5] (adapted from [5] with permission of Frontiers).

**Figure 7 polymers-15-03138-f007:**
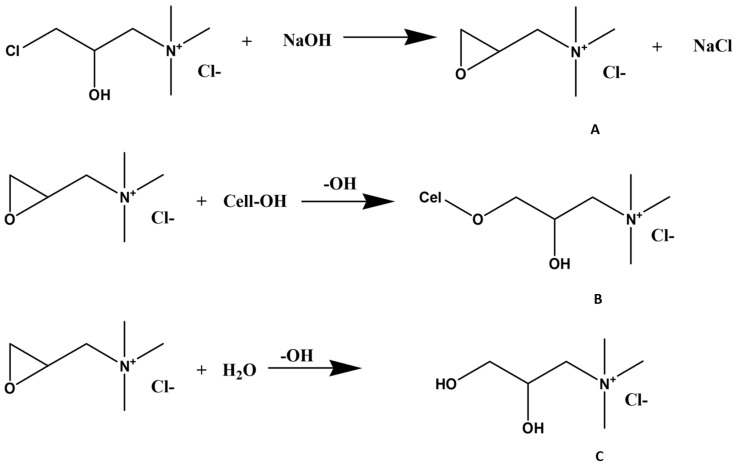
Simplified reaction mechanisms occurring during the cationization of cellulose with CHPTAC under alkaline conditions. Conversion of CHPTAC into EPTAC (**A**); Etherification reaction of cel-lulose with EPTAC (B); Hydrolysis reaction of EPTAC (**C**). Adapted from reference [100] with permission of John Wiley and Sons.

**Figure 8 polymers-15-03138-f008:**
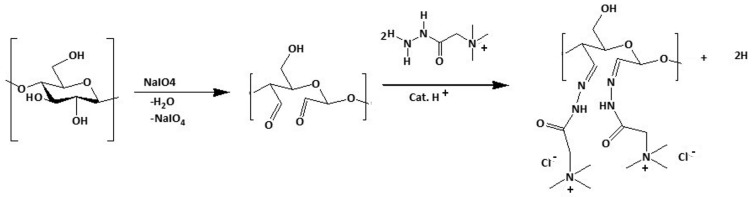
Dual step cationization of cellulose with GT via DAC. Firstly, a periodate oxidation of cellulose to create DAC is conducted, followed by the synthesis of cationic cellulose using Girard’s reagent. Adapted from [106] with permission of Springer.

**Figure 9 polymers-15-03138-f009:**
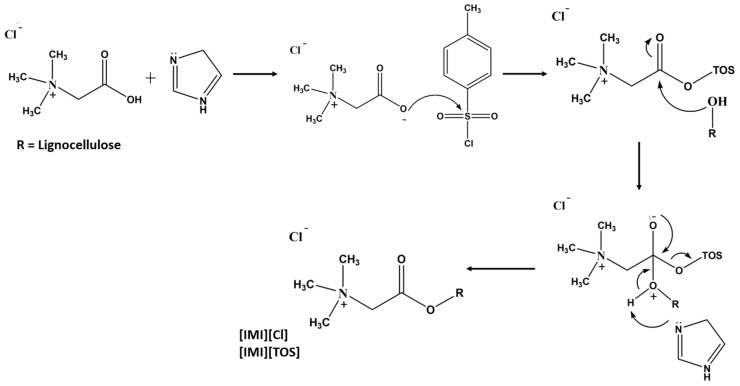
Cationization of cellulose via imidazolium-catalyzed tosylation of betaine hydrochloride. Imidazolium chloride ([IMI][Cl]) and tosylate ([IMI][Tos]) are formed as by-products [108] (adapted from [108] with permission of Elsevier).

**Figure 10 polymers-15-03138-f010:**
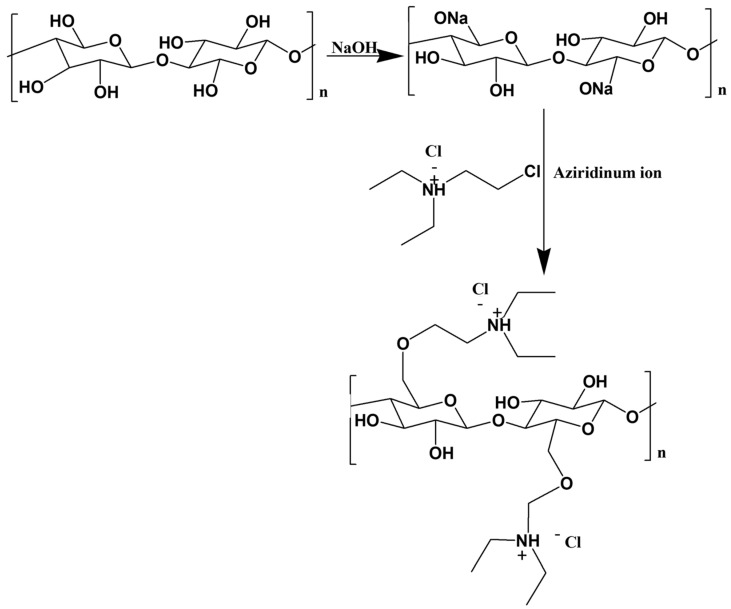
Schematic representation of cellulose quaternization with CEDAC, obtained from the chemical reaction between cellulose and quaternary amine [109] (adapted from reference [109] with permission of Elsevier).

**Figure 11 polymers-15-03138-f011:**
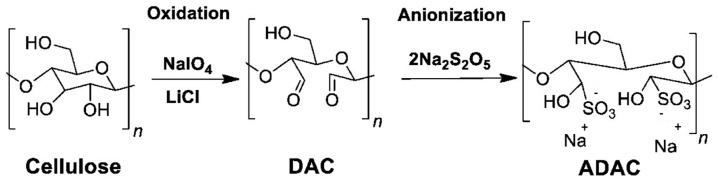
Dual-step anionization of cellulose through sulfonation of DAC (adapted from reference [5] with permission of Frontiers).

**Figure 12 polymers-15-03138-f012:**
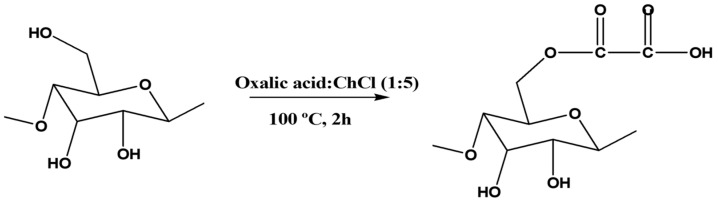
Development of carboxylated cellulose with a DES containing carboxylic acid [111,112].

**Figure 13 polymers-15-03138-f013:**
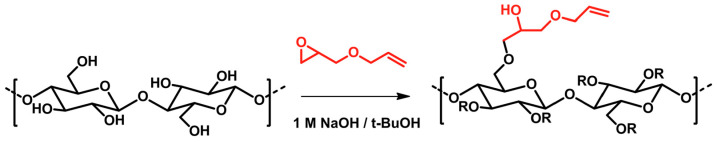
Synthesis of 3-allyloxy-2-hydroxypropyl cellulose [114]. R = H, or 3-allyloxy-2-hydroxypropyl. Reprinted from [114] with permission of Springer.

**Figure 14 polymers-15-03138-f014:**
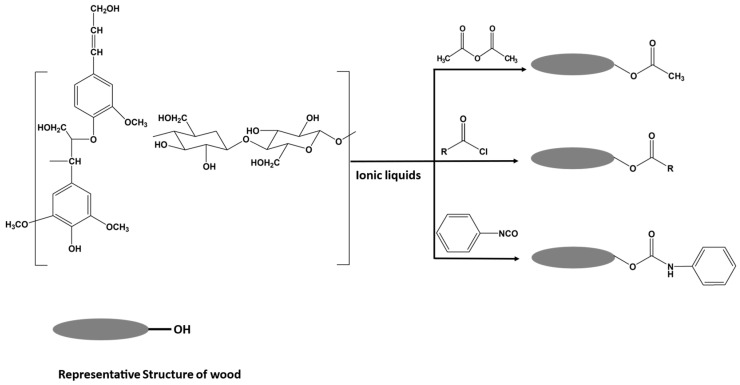
Representative structures of wood components and the homogenous functionalization reactions performed in ILs (adapted from reference [68] with permission of ACS).

**Figure 15 polymers-15-03138-f015:**
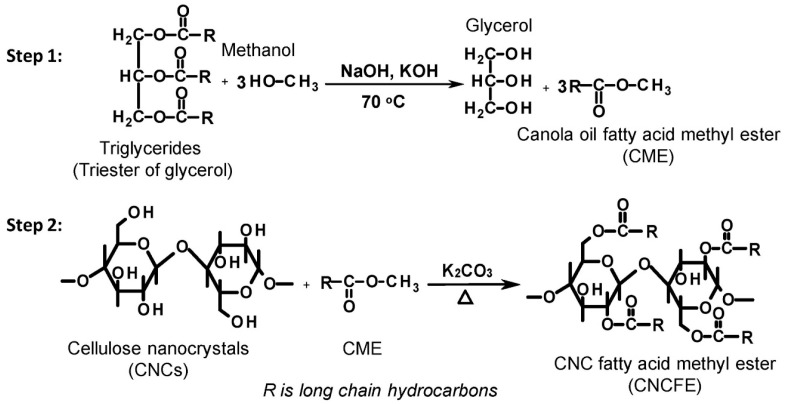
Reaction mechanism for the synthesis of canola oil fatty acid methyl ester (CME) (step 1) and cellulose fatty acid methyl ester (step 2). R represents the long hydrocarbon chain (adapted from reference [117] with permission of Elsevier).

**Figure 16 polymers-15-03138-f016:**
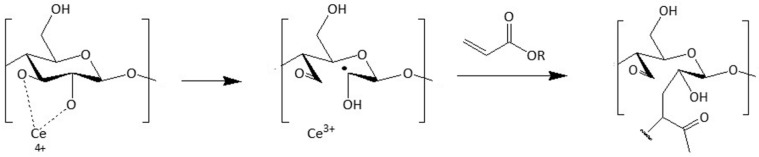
Mechanism of cerium-initiated copolymerization (adapted from references [119,120] with permission of Elsevier).

**Figure 17 polymers-15-03138-f017:**
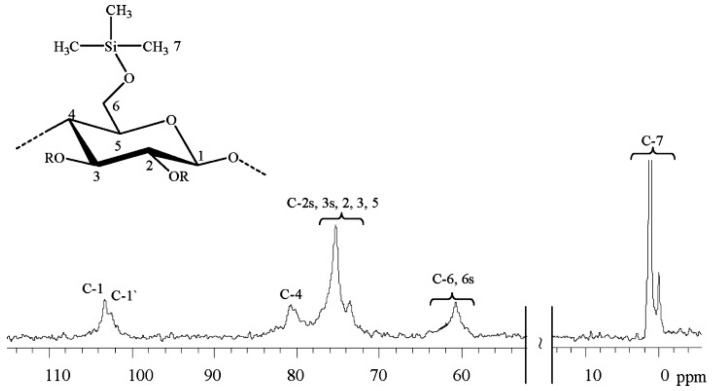
^13^C NMR spectrum of trimethylsilyl cellulose (degree of substitution, DS ≈ 0.43) in DMSO-d_6_, where R means trimethylsilyl group or H according to DS [121]. Reproduced from Reference [124] with permission from Wiley and the Copyright Clearance Center, 2008.

## Data Availability

No new data were created or analyzed in this study. Data sharing is not applicable to this article.

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
