# Peer review of "Eco-Friendly Methods for Extraction and Modification of Cellulose: An Overview"

_polymers, 2023, doi:10.3390/polym15143138_

Round 1

Reviewer 1 Report

Title: On the modification of cellulose from lignocellulosic biomass:  an overview

The paper is  review on Cellulose modifications.  I have some serous concerns as listed below.

1. The author is undecided to focus on a specific topic. The title and abstract are not matching. Reading the abstract to me it seems the focus will be on the extraction of the modified cellulose. looking into green solvents; but that is not the case. Hence a synchronization is necessary.

2. I believe the introduction will highlight the sequence of the review/ overview; what is the story, how that story is compiled and a final message at per with the abstract.

3. The review is not focused, you have to focus on one topic say modification of cellulose, how it is synthesized, characterized, the application, the limitation and the issues.

4. Example cases of applications which is missing, new references are missing, such as:
   a. A recent work of Barabara et al on cellulose purification https://doi.org/10.1016/j.cogsc.2023.100783
   b. It may be as well good to talk about some applications to create functional chemical
https://doi.org/10.1016/j.cattod.2016.02.050

5. My key concern is the connected story. Cellulose this is a vast topic; and there are many reviews.  This review should isolate the content with a  focused area with new references.

6. Cellulose is wide area and there are many reviews in this topic, and the review does even qualify in that level due to the focus.

Author Response

Reviewer #1

“Title: On the modification of cellulose from lignocellulosic biomass:  an overview

The paper is review on Cellulose modifications.  I have some serous concerns as listed below.

1. The author is undecided to focus on a specific topic. The title and abstract are not matching. Reading the abstract to me it seems the focus will be on the extraction of the modified cellulose. looking into green solvents; but that is not the case. Hence a synchronization is necessary.

  1. I believe the introduction will highlight the sequence of the review/ overview; what is the story, how that story is compiled and a final message at per with the abstract.

    3. The review is not focused, you have to focus on one topic say modification of cellulose, how it is synthesized, characterized, the application, the limitation and the issues.”

The authors acknowledge the comments of the reviewer. Based on the reviewer comments we modified the title and the abstract to have a better matching. The focus of the review is the derivatization, but to obtain cellulose fibres in a more sustainable way, we also included some new recent sustainable strategies to extract native cellulose. After cellulose extraction, its chemical modification is discussed. Finally, selected applications involving cellulose derivatives are described.

Additionally, we have also modified the introduction (background) section, to include more information about the main driving force for the present review, as well as, to give details of the organization of the manuscript. The main subsections were ordered in the following sequence: sources of cellulose, green extraction processes, physical-chemical characteristics of cellulose, chemical modification (derivatization) of cellulose and finally selected applications of the prepared cellulose derivatives.

The main message of the present review is that cellulose can be the leading raw material for different applications, promoting the sustainable transition from fossil resources to bio-based materials, while helping the mitigation of the problems potentially arising from climate change effects.

  1. Example cases of applications which is missing, new references are missing, such as:
    a. A recent work of Barabara et al on cellulose purification https://doi.org/10.1016/j.cogsc.2023.100783
    b. It may be as well good to talk about some applications to create functional chemical
    https://doi.org/10.1016/j.cattod.2016.02.050
    The manuscript has been updated with more recent relevant references, such as the first one suggested by the reviewer, even if there were already many recent references in the initial version. Changes are properly highlighted in the revised text.

It is worth mentioning that the second paper suggested (https://doi.org/10.1016/j.cattod.2016.02.050) is not suitable to the present review, because it deals with catalytic hydrogenolysis of the C-O bond of diphenyl ether (a lignin model compound).

  1. My key concern is the connected story. Cellulose this is a vast topic; and there are many reviews.  This review should isolate the content with a focused area with new references.
  2. Cellulose is wide area and there are many reviews in this topic, and the review does even qualify in that level due to the focus.

Following the reviewer comments, we have modified parts of the manuscript to make it more concise and focused. We do agree with the reviewer that cellulose is a vast topic, and there are already many reviews. Nevertheless, this work focusses on the main advances in green methods to extract and modify cellulose which, to our knowledge, have been a less discussed topic. At the end, different applications of the cellulose derivatives were selected and discussed in the text. We believe this structure indeed tells a story from cellulose origin (sources of cellulose) to its extraction and modification, where, depending on the substituting groups, different chemical features arise. Finally, the story in concluded with some selected applications of these cellulose derivatives.

Reviewer 2 Report

Cellulose is the most abundant biopolymer on earth. Many scientists develop innovative ways to improve and tune cellulose properties and grant new functionalities. These strategies typically involve cellulose derivatization by incorporation of cationic, anionic, or hydrophobic functional groups in the cellulose chain. Luís Alves et al review some of cellulose main structural characteristics and properties, chemical modifications and applications. The review is interesting. The author should focus lignocellulosic biomass in the review. Other points of the manuscript also should be improved.

1.   The molecular weight of cellulose from different resource should be added in this review.

2.   Figure 4. Some chemical structure of different cellulose derivatives [75] (adapted from 240 reference [75]). The authors should add more chemical structure of different cellulose derivatives.

3.   7. Cationization, 8. Anionization and 9. Hydrophobic modification should be added in the content of 6 Chemical modification of cellulose.

4.   The authors should add a table in 10. Applications of cellulose derivatives to summarize this part.

5.   “Figure 11: Development of carboxylated cellulose with a DES containing carboxylic acid”. The authors should check this equation in Figure 11 and add the reaction condition of this equation.

6.   The authors should cite more reference within 3 years.

Please carefully check the manuscript for writing and grammar.

Author Response

Reviewer #2

Cellulose is the most abundant biopolymer on earth. Many scientists develop innovative ways to improve and tune cellulose properties and grant new functionalities. These strategies typically involve cellulose derivatization by incorporation of cationic, anionic, or hydrophobic functional groups in the cellulose chain. Luís Alves et al review some of cellulose main structural characteristics and properties, chemical modifications and applications. The review is interesting. The author should focus lignocellulosic biomass in the review. Other points of the manuscript also should be improved.

We are thankful to the reviewer for recognizing the general importance of our manuscript.

  1. The molecular weight of cellulose from different resource should be added in this review.

Following the reviewer suggestion, information on the molecular weight of cellulose from different sources, which was possible to find in literature, was added to the manuscript (Section 2).

  1. Figure 4. Some chemical structure of different cellulose derivatives [75] (adapted from 240 reference [75]). The authors should add more chemical structure of different cellulose derivatives.

We agree with the reviewer suggestion and Figure 4 has been revised accordingly.

  1. 7. Cationization, 8. Anionization and 9. Hydrophobic modification should be added in the content of 6 Chemical modification of cellulose.

We agree with the reviewer. The manuscript has been changed to meet the reviewer concerns. Changes are properly highlighted in the revised text.

  1. The authors should add a table in 10. Applications of cellulose derivatives to summarize this part.

Following the suggestion of the reviewer, a table summarizing the applications of cellulose derivatives (Table 2) was prepared and added in section “7. Applications of cellulose derivatives”.

  1. “Figure 11: Development of carboxylated cellulose with a DES containing carboxylic acid”. The authors should check this equation in Figure 11 and add the reaction condition of this equation.

The figure has been updated and the reaction conditions added (Figure 12).

  1. The authors should cite more reference within 3 years.

It should be highlighted that, in fact, we have tried to give preference for recent (last 3 years) references during the entire manuscript. During this revision process, new references were identified and added. Please note that we have decided to maintain some older works either due to their overall relevance in the field and/or if no more recent references were available.

Reviewer 3 Report

Comments and Suggestions for Authors

I have read the manuscript: On the modification of cellulose from lignocellulosic biomass: an overview

My comment and suggestions are presented below:

1.      The title of the manuscript isn't sound and therefore, should be improved an example can be: State of the art of ecofriendly methods used for improving cellulose proprieties, or other title…

2.      The abstract is not sufficiently developed.

3.      Introduction part where is? All articles have an introduction whether it is a review or an article, etc. Please development the subject, you don’t present enough information regarding the state of the art of the subject studied.

4.      Could you give more detail about the importance of this Review and please develop the main goal of this review?

5.      Please, make a figure for green methods for cellulose extraction with advantages and disadvantages (at point 3, page 3).

6.      You have many figures that are written as: adapted from the reference, because do not give other information, you must have the permission of the Journal.

7.      Please, try to separate and highlight each sub-point better. It was difficult for me to follow quantitatively all the information presented. Therefore, the article needs to be better organized.

8.      Some subtitles are given in a general way e.g. cationalization. Please define the subtitle correctly in order to represent the information presented.

9.      XRD, FTIR, SEM, TEM images in sub points 7, 8 and 9 you have not added to highlight the morphological transformations of cellulose.

10.  The basic rule when choosing a Journal is necessary to consult the writing requirements. Therefore, please read them and see what you did not respect.

11.  At point 10. Applications of cellulose derivatives, you presented them one after the other without making a scheme of their applications (e.g. obtaining of biomaterials, etc.)

Author Response

Reviewer #3

I have read the manuscript: On the modification of cellulose from lignocellulosic biomass: an overview.

My comment and suggestions are presented below:

  1. The title of the manuscript isn't sound and therefore, should be improved an example can be: State of the art of ecofriendly methods for improving cellulose proprieties, or other title…

We thank the reviewer comment. Following the reviewer suggestion, the title has been modified. The new title is: “Ecofriendly methods for extraction and modification of cellulose: an overview”.

  1. The abstract is not sufficiently developed.

Thanks for your observation, which has driven us to improve the abstract. The changes are properly highlighted in the revised text.

  1. Introduction part where is? All articles have an introduction whether it is a review or an article, etc. Please development the subject, you don’t present enough information regarding the state of the art of the subject studied.

We agree with the reviewer. This section Introduction (named Background) has been improved and rephrased, and the novelty of the works better highlighted in the text.

  1. Could you give more detail about the importance of this Review and please develop the main goal of this review?

According to the reviewer concern, we added more details about the novelty and importance of the present review. The text added is properly highlighted in the revised manuscript.

  1. Please, make a figure for green methods for cellulose extraction with advantages and disadvantages (at point 3, page 3).

A new Figure was added to the document (Figure 2), containing the main advantages and disadvantages of green methods for cellulose extractions.

  1. You have many figures that are written as: adapted from the reference, because do not give other information, you must have the permission of the Journal.

We updated the information regarding the figures adapted. Note that all the required permissions were obtained from the different journals.

  1. Please, try to separate and highlight each sub-point better. It was difficult for me to follow quantitatively all the information presented. Therefore, the article needs to be better organized.

The manuscript was reorganized to meet the reviewer concerns. For example, points 7, 8 and 9, were included in the sub-section “Chemical modification of cellulose”, while in the applications section, a table was added, summarizing some of the relevant applications.

  1. Some subtitles are given in a general way e.g. cationalization. Please define the subtitle correctly in order to represent the information presented.

The subtitles were modified to better represent the information disclosed in each section. Sections 7, 8 and 9 were inserted in Section 6, as subtitles 6.1, 6.2 and 6.3, which enables an easier understanding of the Sections and its content. The subtitles “Cationization”, “Anionization” and “Hydrophobic modification” are now subsections of the Section 6 “Chemical modification of cellulose”.

  1. XRD, FTIR, SEM, TEM images in sub points 7, 8 and 9 you have not added to highlight the morphological transformations of cellulose.

We understand the reviewer point, but in this paper we have decided to mainly focus on the different reactions and mechanisms involving the modification of cellulose. Although very interesting and important, the characterization of the different cellulose derivatives by different methods would broaden the scope of the present paper and dilute the main message of it which essentially regards the principal green strategies to extract and modify cellulose. Also, this information is not always available in the manuscripts dealing with cellulose modification.

  1. The basic rule when choosing a Journal is necessary to consult the writing requirements. Therefore, please read them and see what you did not respect.

We thank the reviewer comment. The authors guidelines were carefully re-analysed, and we believe all journal requirements are being followed.

  1. At point 10. Applications of cellulose derivatives, you presented them one after the other without making a scheme of their applications (e.g. obtaining of biomaterials, etc.

Following the suggestion of the reviewer, a table summarizing selected applications of cellulose derivatives (Table 2) was prepared and added to the manuscript in the section “7. Applications of cellulose derivatives”.

Round 2

Reviewer 1 Report

The authors have responded to all my concerns and therefore I am happy to recommend publication a is  

Minor editing is required

Reviewer 3 Report

The authors have addressed the majority of my questions, I recommend the paper for publication in Journal.